# Where Did the Gap Go?
# Reassessing the Long-Range Graph Benchmark

**Jan Tönshoff**                                                  *toenshoff@informatik.rwth-aachen.de*
*RWTH Aachen University*

**Martin Ritzert**                                              *ritzert@informatik.uni-goettingen.de*
*Georg-August-Universität Göttingen*

**Eran Rosenbluth**                                          *rosenbluth@informatik.rwth-aachen.de*
*RWTH Aachen University*

**Martin Grohe**                                                    *grohe@informatik.rwth-aachen.de*
*RWTH Aachen University*

**Reviewed on OpenReview:** *https: // openreview. net/ forum? id= NmOWX86sKv*

## Abstract

The recent Long-Range Graph Benchmark (LRGB, Dwivedi et al. 2022) introduced a set of graph learning tasks strongly dependent on long-range interaction between vertices. Empirical evidence suggests that on these tasks Graph Transformers significantly outperform Message Passing GNNs (MPGNNs). In this paper, we carefully reevaluate multiple MPGNN baselines as well as the Graph Transformer GPS (Rampáek et al. 2022) on LRGB. Through a rigorous empirical analysis, we demonstrate that the reported performance gap is overestimated due to suboptimal hyperparameter choices. It is noteworthy that across multiple datasets the performance gap completely vanishes after basic hyperparameter optimization. In addition, we discuss the impact of lacking feature normalization for LRGB's vision datasets and highlight a spurious implementation of LRGB's link prediction metric. The principal aim of our paper is to establish a higher standard of empirical rigor within the graph machine learning community.

## 1 Introduction

Graph Transformers (GTs) have recently emerged as a popular alternative to conventional Message Passing Graph Neural Networks (MPGNNs) which dominated deep learning on graphs for years. A central premise underlying GTs is their ability to model long-range interactions between vertices through a global attention mechanism. This could give GTs an advantage on tasks where MPGNNs may be limited through phenomenons like over-smoothing, over-squashing, and under-reaching, thereby justifying the significant runtime overhead of self-attention.

The Long-Range Graph Benchmark (LRGB) has been introduced by Dwivedi et al. (2022) as a collection of five datasets:

- *Peptides-func* and *Peptides-struct* are graph-level classification and regression tasks, respectively. Their aim is to predict various properties of peptides which are modelled as molecular graphs.
- *PascalVOC-SP* and *COCO-SP* model semantic image segmentation as a node-classification task on superpixel graphs.
- *PCQM-Contact* is a link prediction task on molecular graphs. The task is to predict pairs of vertices which are distant in the graph but in contact in 3D space.

| Dataset | Peptides-Func | Peptides-Struct | PascalVOC-SP | COCO-SP | PCQM-Contact |
|---|---|---|---|---|---|
| Domain | Chemistry | Chemistry | Vision | Vision | Chemistry |
| Task | Graph Cls. | Graph Reg. | Node Cls. | Node Cls. | Link Pred. |
| #Graphs | 15,535 | 15,535 | 11,355 | 123,286 | 529,434 |
| Avg. Nodes | 150.94 | 150.94 | 479.40 | 476.88 | 30.14 |
| Avg. Diameter | 56.99 | 56.99 | 27.62 | 27.39 | 9.86 |

Table 1: Overview of the five Long-Range Graph Benchmark datasets.

Table 1 provides the main statistics of each dataset. The five LRGB datasets contain graphs that are larger and of higher diameter than those used in prior datasets for similar prediction tasks. A core hypothesis of LRGB is that the chosen learning tasks depend strongly on long-range dependencies between vertices and are therefore suitable for comparing architectures that specifically aim to capture such dependencies, such as GTs.

The experiments provided by Dwivedi et al. (2022) report a strong performance advantage of GTs over the MPGNN architectures GCN (Kipf & Welling, 2017), GINE (Hu et al., 2020b), and GatedGCN (Bresson & Laurent, 2017), in accordance with the expectations. Subsequently, GPS (Rampášek et al., 2022) reached similar conclusions on LRGB. We note that these two works are strongly related and built on a shared code base. Newer research on GTs (see Section 1.1) is commonly based on forks of this code base and often cites the baseline performance reported by Dwivedi et al. (2022) to represent MPGNNs.

Our contribution is three-fold[1]: First, we show that the three MPGNN baselines GCN, GINE, and GatedGCN all profit massively from further hyperparameter tuning, reducing and even closing the gap to graph transformers on multiple datasets. In fact, GCN yields state-of-the-art results on Peptides-Struct, surpassing several newer graph transformers. On this dataset in particular, most of the performance boost is due to a multi-layer prediction head instead of a linear one, again highlighting the importance of hyperparameters. Second, we show that on the vision datasets PascalVOC-SP and COCO-SP normalization of the input features is highly beneficial. We argue that, as in the vision domain, feature normalization should be the default setting. Third and last we take a closer look at the MRR metric used to evaluate PCQM-Contact. There, we demonstrate different filtering strategies have a major impact on the results and must be implemented exactly to specification to facilitate reliable comparisons.

## 1.1 Related Work

Our primary focus are the commonly used MPGNNs GCN (Kipf & Welling, 2017), GINE (Hu et al., 2020b), and GatedGCN (Bresson & Laurent, 2017) as well as the graph transformer GPS (Rampášek et al., 2022). GCN performs simple message passing without edge features. GINE extends Graph Isomorphism Networks (GIN) (Xu et al., 2019) by additionally incorporating edge features. GatedGCN also incorporates edge features to compute a channel-wise 'importance weighting' of messages. These three architectures are all standard MPGNNs (Gilmer et al., 2017) where the depth of the receptive field of each vertex is equal to the number of message-passing layers. The graph transformer GPS applies message-passing and self-attention (Vaswani et al., 2017) in parallel and combines the results in each layer. With this additional global information exchange facilitated through self-attention, each vertex generally has an unbounded receptive field and can directly aggregate information from the entire graph. This should in principle help graph transformers like GPS capture long-range dependencies on graphs.

There are many more MPGNN architectures (Hamilton et al., 2017; Xu et al., 2018; Chen et al., 2020; Corso et al., 2020), as well as graph transformers (Dwivedi & Bresson, 2020; Ying et al., 2021; Kreuzer et al., 2021; Shi et al., 2020; Park et al., 2022; Weis et al., 2021; Rampášek et al., 2022; Shirzad et al., 2023; Kim et al., 2022; Ma et al., 2023; He et al., 2023), see also the survey by Min et al. (2022). Many newer graph transformer architectures have reported results on LRGB datasets, including Exphormer (Shirzad

---

[1]Source code: `https://github.com/toenshoff/LRGB`

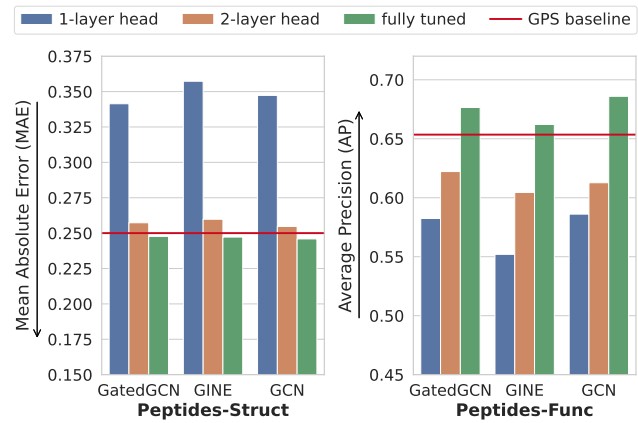

| | Method | PEPTIDES-FUNC | | PEPTIDES-STRUCT | |
|---|---|---|---|---|---|
| | | Test AP ↑ | rel imp | Test MAE ↓ | rel imp |
| LRGB | GCN | 0.5930 ± 0.0023 | | 0.3496 ± 0.0013 | |
| | GINE | 0.5498 ± 0.0079 | | 0.3547 ± 0.0045 | |
| | GatedGCN | 0.6069 ± 0.0035 | | 0.3357 ± 0.0006 | |
| | Transformer | 0.6326 ± 0.0126 | | 0.2529 ± 0.0016 | |
| | SAN | 0.6439 ± 0.0075 | | 0.2545 ± 0.0012 | |
| | GPS | 0.6535 ± 0.0041 | | 0.2500 ± 0.0005 | |
| Ours | GCN | 0.6860 ± 0.0050 | +16% | **0.2460 ± 0.0007** | +30% |
| | GINE | 0.6621 ± 0.0067 | +20% | 0.2473 ± 0.0017 | +30% |
| | GatedGCN | 0.6765 ± 0.0047 | +11% | 0.2477 ± 0.0009 | +26% |
| | GPS | 0.6534 ± 0.0091 | ± 0% | 0.2509 ± 0.0014 | ± 0% |
| Others | CRaWl | 0.7074 ± 0.0032 | | 0.2506 ± 0.0022 | |
| | DRew | **0.7150 ± 0.0044** | | 0.2536 ± 0.0015 | |
| | Exphormer | 0.6527 ± 0.0043 | | 0.2481 ± 0.0007 | |
| | GRIT | 0.6988 ± 0.0082 | | **0.2460 ± 0.0012** | |
| | Graph ViT | 0.6942 ± 0.0075 | | **0.2449 ± 0.0016** | |
| | G-MLPMixer | 0.6921 ± 0.0054 | | 0.2475 ± 0.0015 | |

(a) Previous and updated results on the peptides datasets. Best results (within stdev) in **bold**.

(b) Exchanging the linear prediction head by an MLP accounts for most of the additional performance of all three MPGNNs, especially on Peptides-Struct

Figure 1: On both Peptides datasets, all three MPGNNs surpass GPS. On Peptides-Struct a basic GCN model even achieves SOTA results.

et al., 2023), GRIT (Ma et al., 2023) and Graph ViT / GraphMLPMixer (He et al., 2023). Several other architectures not based on transformers have also been evaluated on LRGB, including CRaWl (Tönshoff et al., 2023), DRew (Gutteridge et al., 2023) and Virtual Nodes (Gilmer et al., 2017; Cai et al., 2023). In general, long-range interactions are typically handled through GNNs with many layers (e.g. Li et al., 2020), or the addition of global information exchange through graph transformers and virtual nodes (Gilmer et al., 2017). Finally, we do see a connection of our work to graph learning benchmarking projects (Dwivedi et al., 2020; Hu et al., 2020a) that also advocate for rigorous testing of graph learning architectures.

## 2 Concerns

**Hyperparameters** In this paper, we argue that the results reported by Dwivedi et al. (2022) are not representative of MPGNNs and suffer from suboptimal hyperparameters. We provide new results for the same MPGNN architectures that are obtained after a basic hyperparameter sweep. We tune the main hyperparameters (such as depth, dropout rate, ...) in pre-defined ranges while strictly adhering to the official 500k parameter budget. The exact hyperparameter ranges and all final configurations are provided in Appendix A.1. In particular, we looked at networks with 6 to 10 layers, varied the number of layers in the prediction head from 1 to 3 (which turned out to be very relevant), and also considered the dropout and learning rate of the network. In addition to those "standard" hyperparameters, we also view the usage of positional or structural encoding (none / LapPE (Dwivedi & Bresson, 2020) / RWSE (Dwivedi et al., 2021)) as a hyperparameter that is tuned for each method. As a point of reference, we reevalute GPS in an identical manner, additionally tuning the inner MPGNN and the normalization strategy, and also achieve significantly improved results on three datasets with this Graph Transformer. The results reported for GPS may therefore also be subject to suboptimal configurations.

**Feature Normalization** The vision datasets PascalVOC-SP and COCO-SP have multi-dimensional node and edge features with values spanning different orders of magnitude for different feature channels. Passing this input to a neural network without channel-wise normalization can cause poorly conditioned activations. While feature normalization is standard practice in deep learning and computer vision in particular, neither Dwivedi et al. (2022) nor any subsequent works using LRGB utilize it, except CRaWl (Tönshoff et al., 2023). We apply channel-wise linear normalization to all input features as visualized in Figure 2a. We show that all models (baselines and GPS) profit from it in an ablation in Figure 2b.

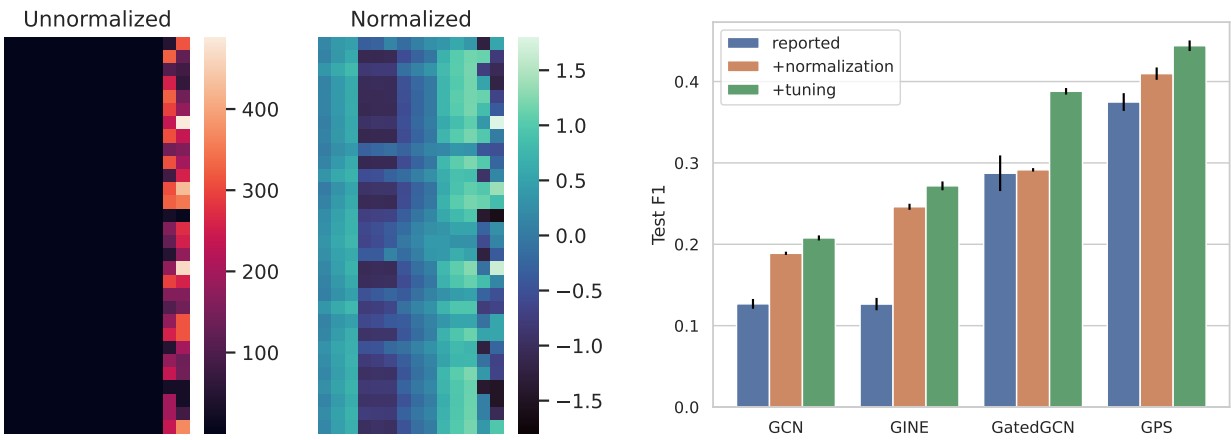

(a) Vertex feature matrices ($|V| \times 14$) on PascalVOC-SP before and after channel-wise normalization.

(b) The effects of feature normalization and hyperparameter tuning on PascalVOC-SP.

Figure 2: On PascalVOC-SP and COCO-SP feature normalization and further tuning improves performance across all compared methods.

**Link Prediction Metrics**    The evaluation metric on the link-prediction dataset PCQM-Contact (Dwivedi et al., 2022) is the Mean Reciprocal Rank (MRR) in a *filtered* setting, as defined by Bordes et al. (2013). For predicted edge scores the MRR measures how a given true edge $(h, t)$ is ranked compared to all possible candidate edges $(h, x)$ of the same head. As there might be multiple true tails $t$ for each head $h$, the *filtered* MRR removes those other true tails (false negatives) from the list of candidates before computing the metric. This filtering avoids erroneously low MRR values due to the model preferring other true edges and is common in link-prediction tasks. Even though Dwivedi et al. (2022) explicitly define the metric to be the filtered MRR, the provided code computes the *raw* MRR, i.e. keeping other true tails in the list. We report results on PCQM-Contact in a corrected filtered setting. We additionally provide results with an extended filtering procedure where self-loops of the form $(h, h)$ are also removed from the set of candidates since these are semantically meaningless and never positive. This is impactful as the scoring function used by Dwivedi et al. (2022) is based on a symmetric dot-product and therefore exhibits a strong bias towards self-loops.

## 3    Experiments

**Peptides-Func and Peptides-Struct**    Table 1a provides the results obtained on the test splits of the Peptides-Func and Peptides-Struct. For the MPGNN baselines, we observe considerable improvements on both datasets as all three MPGNNs outperform GPS after tuning. The average precision on Peptides-Func increased relatively by around 10% to 20%. GCN achieves a score of 68.60%, which is competitive with newer GTs such as GRIT or Graph ViT. The improvement on Peptides-Struct is even more significant with a relative reduction of the MAE of 30%, fully closing the gap to recently proposed GTs. Surprisingly, a simple GCN is all you need to match the best-known results on Peptides-Struct. The results for GPS effectively stayed the same as in the original paper (Rampášek et al., 2022). Those values thus seem to be representative for GPS.

We observed that the key hyperparameter underlying the improvements of all three MPGNNs is the depth of the prediction head. To show this Figure 1b contains an ablation where we exchanged the linear prediction head configured by Dwivedi et al. (2022) with a 2-layer perceptron, keeping all other hyperparameters the same. While the benefit on Peptides-Func is considerable and highly significant, on Peptides-Struct the head depth accounts for almost the complete performance gap between MPGNNs and GTs. GPS' performance with linear and deeper prediction heads is largely unchanged. For example, our GPS configurations in Table 1a use a 2-layer prediction head. Our results indicate that the prediction targets of both datasets appear to depend non-linearly on global graph information. In this case, MPGNNs with linear prediction heads are

| Method | PascalVOC-SP Test F1 ↑ | Rel Imp | COCO-SP Test F1 ↑ | Rel Imp |
|---|---|---|---|---|
| **LRGB** | | | | |
| GCN | 0.1268 ± 0.0060* | | 0.0841 ± 0.0010* | |
| GINE | 0.1265 ± 0.0076* | | 0.1339 ± 0.0044* | |
| GatedGCN | 0.2873 ± 0.0219* | | 0.2641 ± 0.0045* | |
| Transformer | 0.2694 ± 0.0098* | | 0.2618 ± 0.0031* | |
| SAN | 0.3230 ± 0.0039* | | 0.2592 ± 0.0158* | |
| GPS | 0.3748 ± 0.0109* | | 0.3412 ± 0.0044* | |
| **Ours** | | | | |
| GCN | 0.2078 ± 0.0031 | +64% | 0.1338 ± 0.0007 | +59% |
| GINE | 0.2718 ± 0.0054 | +115% | 0.2125 ± 0.0009 | +59% |
| GatedGCN | 0.3880 ± 0.0040 | +35% | 0.2922 ± 0.0018 | +11% |
| GPS | 0.4440 ± 0.0065 | +18% | **0.3884 ± 0.0055** | +13% |
| **Others** | | | | |
| CRaWL | **0.4588 ± 0.0079** | - | | |
| DRew | 0.3314 ± 0.0024* | - | | |
| Exphormer | 0.3960 ± 0.0027* | | 0.3430 ±0.0008* | |

(a) Tuning results on vision datasets PascalVOC-SP and COCO-SP. *No normalization used.

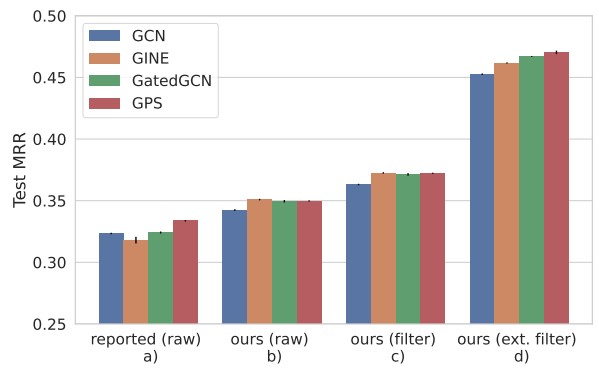

(b) Results on PCQM-Contact. For our own models we provide the MRR scores with varying levels of filtering.

unable to model the target function. Graph Transformers are not as sensitive to linear prediction heads, since each layer can process global graph information with a deep feed-forward network. However, we would argue that switching to a deep predictive head represents a simpler and computationally cheaper solution to the same issue.

**PascalVOC-SP and COCO-SP**    Table 3a provides the results obtained on the test splits of the superpixel datasets PascalVOC-SP and COCO-SP. We observe significant improvements for all evaluated methods. On PascalVOC-SP the F1 score of GatedGCN increases to 38.80% which exceeds the original performance reported for GPS by Rampášek et al. (2022). GPS also improves significantly to 44.40% F1. This is only one percentage point below the results achieved by CRaWL, which currently is the only reported result with normalized features. The previously large performance gap between GPS and CRaWL is therefore primarily explained by GPS processing raw input signals. On COCO-SP, we observe similar results. Here GPS sets a new state-of-the-art F1 score of 38.84%.

Note that these improvements are achieved entirely through data normalization and hyperparameter tuning. Figure 2b provides an ablation on the individual effect of normalization. We train intermediate models with configurations identical to those used by Dwivedi et al. (2022) and Rampášek et al. (2022), but with feature normalization. For GatedGCN we observe a slight performance increase but a large reduction in the variance across random seeds. For the remaining methods, including GPS, normalization of node and edge features already accounts for at least half of the observed performance gain, emphasizing its importance in practice.

**PCQM-Contact**    Figure 3b plots the MRR scores obtained on the test split with various evaluation settings as described in the link prediction paragraph of Section 2. First, we provide the results originally reported for LRGB in the literature (a). Recall that these values are obtained in a raw setting with false negatives present. We then provide results obtained after training our own model with new hyperparameters (chosen based on the raw MRR) in b). We still use the raw MRR for evaluation in b) to measure the impact of hyperparameter tuning. Tuning yields an absolute improvement of around 3%. The previously reported slight performance edge of GPS is not observable in this setting after tuning.

In subplot c) we measure the MRR of our models in the filtered setting. Note that these values are based on the exact same predictions as in b), but false negatives are removed. The measured MRR increases by roughly 3% when compared to the raw setting. This shift could erroneously be interpreted as a significant improvement when comparing to literature values obtained in a raw setting. In d) we evaluate our models (still using the same predictions) in an extended filtered setting where we additionally remove self-loops from the candidate pool. Compared to the filtered MRR in c) the MRR metric increases by about 10 percentage points, indicating that self-loops strongly affect the results. Note that in d) GPS again slightly outperforms the MPGNN baselines, in contrast to b) and c). This means that GPS' predictions seem to suffer overproportionally when self-loops are not filtered. Therefore, the specific choice of how negative

samples are filtered on PCQM-Contact can directly affect the ranking of compared methods and must be considered and implemented with care.

## 4  Conclusion

In our experiments, we observed considerable performance gains for all three MPGNN baselines. First, this indicates that extensive baseline tuning is important for properly assessing one's own method, especially on relatively recent datasets. Second, only on the two superpixel datasets do graph transformers exhibit clear performance benefits against MPGNNs, indicating that either there are ways to solve the other tasks without long-range interactions or graph transformers are not inherently better at exploiting such long-range dependencies. Evaluating this further appears to be a promising direction for future research. In addition, we would invite a discussion on the best-suited link prediction metric on PCQM-Contact.

Going forward, additional benchmark datasets that test for LRI would also be highly useful to further evaluate existing and future architectures. A stronger focus on synthetic tasks may be helpful to overcome the caveats of the LRGB datasets, where it is unclear how strongly the proposed real-world tasks truly depend on LRI. Classical algorithmic tasks such as path-finding or vertex coloring could be used to define target functions that, by construction, require LRI on synthetic high-diameter graphs.

To evaluate the potential virtues of graph transformers it may further be important to evaluate models on datasets that are large-scale, rather than long-range. When vision transformers were initially shown to outperform CNNs on certain vision tasks, this was only achieved when pre-training on massive datasets with over 100 Million images (Dosovitskiy et al., 2021). While more data-efficient vision transformers have since been developed (Touvron et al., 2021), the general consensus that transforms are "data hungry" prevails across domains. Many popular benchmark datasets in graph learning are comparatively small and often contain less than 100K graphs. These datasets may therefore simply be to small to observe potential scaling benefits that transformers have demonstrated in other domains. Recent work has begun to investigate the scaling of MPGNNs and graph transformers to new large-scale datasets (Beaini et al., 2024; Sypetkowski et al., 2024). This research direction may open up new perspectives on the strengths and weaknesses of graph transformers.

### Acknowledgments

This work was supported by the German Research Foundation (DFG) under grants GR 1492/16-1 and KI 2348/1-1. The project also received funding from the European Research Council (ERC) under the European Unions Horizon Europe research and innovation program (Grant Agreement No. 101041669).

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

# A  Experiment Details

## A.1  Hyperparameters

In the following we describe our methodology for tuning hyperparameters on the LRGB datasets. We did not conduct a dense grid search, since this would be infeasible for all methods and datasets. Instead we perform a "linear" hyperparameter search. We start from a empricially chosen default config and tune each hyperparameter individually within a fixed range. Afterwards, we also evaluate the configuration obtained by combining the best choices of every hyperparameter. From all tried configurations we then select the one with the best validation performance as our final setting. For this hyperparameter sweep, we resorted to a single run per configuration and for the larger datasets slightly reduced the number of epochs. For the final evaluations runs we average results across four different random seeds as specified by the LRGB dataset.

Overall, we tried to incorporate the most important hyperparameters which we selected to be dropout, model depth, prediction head depth, learning rate, and the used positional or structural encoding. For GPS we additionally evaluated the internal MPGNN (but only between GCN and GatedGCN) and whether to use BatchNorm or LayerNorm. Thus, our hyperparamters and ranges were as follows:

- Dropout [0, 0.1, 0.2], default 0.1

- Depth [6,8,10], default 8. The hidden dimension is chosen to stay within a hard limit of 500k parameters

- learning rate [0.001, 0.0005, 0.0001], default 0.001

- Head depth [1,2,3], default 2

- Encoding [none, LapPE, RWSE] default none

- Internal MPGNN [GCN, GatedGCN], default GatedGCN (only for GPS)

- Normalization [BatchNorm, LayerNorm] default BatchNorm (only for GPS)

On the larger datasets PCQM-Contact and COCO we reduce the hyperparameters budget slightly for efficiency. There, we did not tune the learning rate (it had been 0.001 in every single other case) and omitted a dropout rate of 0. We note that the tuning procedure used here is relatively simple and not exhaustive. The ranges we searched are rather limited, especially in terms of network depth, and could be expanded in the future. Tables 2 to 6 provide all final model configurations after tuning. Table 7 provides the final performance on all datasets and Figure 4 shows the results from all tuning runs on Peptides Struct and PascalVOC. From the tuning results we observed comparable stability of MPGNN and GT architectures with respect to their hyperparameters.

We make some additional setup changes based on preliminary experiments. All models are trained with an AdamW optimizer using a cosine annealing learning rate schedule and linear warmup. This differs from Dwivedi et al. (2022), who optimized the MPGNN models with a "Reduce on Plateau" schedule and instead matches the learning rate schedule of GPS Rampášek et al. (2022). We set the weight decay to 0.0 in all five datasets and switch to slightly larger batch sizes to speed up convergence. We also choose GeLU Hendrycks & Gimpel (2016) as our default activation function. Furthermore, we change the prediction head for graph-level tasks such that all hidden layers have the same hidden dimension as the GNN itself. These were previously configured to become more narrow with depth, but we could not observe any clear benefit from this design choice. Last, all MPGNN models use proper skip connections which go around the entire GNN layer. The original LRGB results use an implementation of GCN as provided by GraphGym You et al. (2020). The skip connections in this implementation do not skip the actual non-linearity at the end of each GCN layer, possibly hindering the flow of gradients. We reimplement GCN with skip connections that go around the non-linearity. Note that these additional tweaks are **not** used in our ablation studies in Figure 1b and Figure 2b when training the intermediate models where we only change the head depth and normalization, respectively. There, we use identical model configurations to those used in the literature.

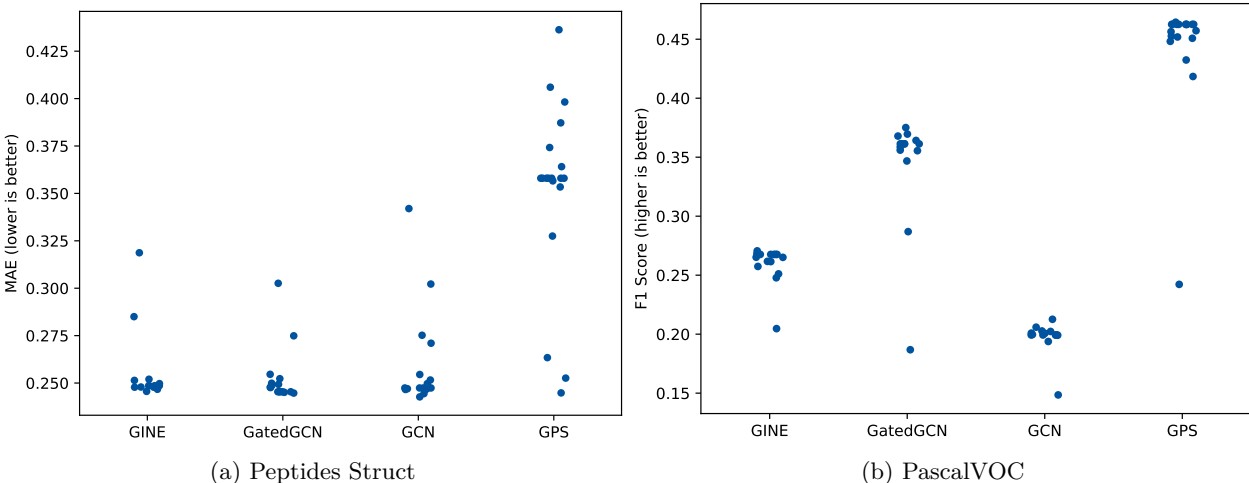

(a) Peptides Struct

(b) PascalVOC

Figure 4: Validation results from all tuning runs on Peptides Struct and PascalVOC. Most results cluster tightly around the best-performing variant of the model.

## A.2  Feature Normalization

On PascalVOC-SP and COCO-SP we apply channel-wise normalisation to the node and edge features. For each dataset, we compute the channel-wise mean $\mu \in \mathbb{R}^d$ and standard deviation $\sigma \in \mathbb{R}^d$ on the train split. Here, $d$ is the feature dimension. Each feature vector $x \in \mathbb{R}^d$ is then normalized linearly before beigng passed to the model:

$$\tilde{x}_i = \frac{x_i - \mu_i}{\sigma_i}$$

Table 2: Hyperparameters on Peptides-Func

|            | GCN   | GINE  | GatedGCN | GPS       |
|------------|-------|-------|----------|-----------|
| lr         | 0.001 | 0.001 | 0.001    | 0.001     |
| dropout    | 0.1   | 0.1   | 0.1      | 0.1       |
| #layers    | 6     | 8     | 10       | 6         |
| hidden dim.| 235   | 160   | 95       | 76        |
| head depth | 3     | 3     | 3        | 2         |
| PE/SE      | RWSE  | RWSE  | RWSE     | LapPE     |
| batch size | 200   | 200   | 200      | 200       |
| #epochs    | 250   | 250   | 250      | 250       |
| norm       | -     | -     | -        | BatchNorm |
| MPNN       | -     | -     | -        | GatedGCN  |
| #Param.    | 486k  | 491k  | 493k     | 479k      |

Table 3: Hyperparameters on Peptides-Struct.

|            | GCN   | GINE  | GatedGCN | GPS       |
|------------|-------|-------|----------|-----------|
| lr         | 0.001 | 0.001 | 0.001    | 0.001     |
| dropout    | 0.1   | 0.1   | 0.1      | 0.1       |
| #layers    | 6     | 10    | 8        | 8         |
| hidden dim.| 235   | 145   | 100      | 64        |
| head depth | 3     | 3     | 3        | 2         |
| PE/SE      | LapPE | LapPE | LapPE    | LapPE     |
| batch size | 200   | 200   | 200      | 200       |
| #epochs    | 250   | 250   | 250      | 250       |
| norm       | -     | -     | -        | BatchNorm |
| MPNN       | -     | -     | -        | GatedGCN  |
| #Param.    | 488k  | 492k  | 445k     | 452k      |

Table 4: Hyperparameters on PascalVOC-SP.

|            | GCN   | GINE  | GatedGCN | GPS       |
|------------|-------|-------|----------|-----------|
| lr         | 0.001 | 0.001 | 0.001    | 0.001     |
| dropout    | 0.0   | 0.2   | 0.2      | 0.1       |
| #layers    | 10    | 10    | 10       | 8         |
| hidden dim.| 200   | 145   | 95       | 68        |
| head depth | 3     | 2     | 2        | 2         |
| PE/SE      | RWSE  | none  | none     | LapPE     |
| batch size | 50    | 50    | 50       | 50        |
| #epochs    | 200   | 200   | 200      | 200       |
| norm       | -     | -     | -        | BatchNorm |
| MPNN       | -     | -     | -        | GatedGCN  |
| #Param.    | 490k  | 450k  | 473k     | 501k      |

Table 5: Hyperparameters on COCO-SP.

|            | GCN   | GINE  | GatedGCN | GPS       |
|------------|-------|-------|----------|-----------|
| lr         | 0.001 | 0.001 | 0.001    | 0.001     |
| dropout    | 0.1   | 0.1   | 0.1      | 0.1       |
| #layers    | 6     | 6     | 8        | 8         |
| hidden dim.| 280   | 195   | 105      | 68        |
| head depth | 1     | 1     | 1        | 1         |
| PE/SE      | none  | none  | none     | none      |
| batch size | 50    | 50    | 50       | 50        |
| #epochs    | 200   | 200   | 200      | 200       |
| norm       | -     | -     | -        | LayerNorm |
| MPNN       | -     | -     | -        | GatedGCN  |
| #Param.    | 500k  | 478k  | 459k     | 500k      |

Table 6: Hyperparameters on PCQM-Contact.

|  | GCN | GINE | GatedGCN | GPS |
|---|---|---|---|---|
| lr | 0.001 | 0.001 | 0.001 | 0.001 |
| dropout | 0.1 | 0.1 | 0.1 | 0.0 |
| #layers | 8 | 8 | 8 | 6 |
| hidden dim. | 215 | 160 | 105 | 76 |
| head depth | 1 | 1 | 1 | 1 |
| PE/SE | LapPE | LapPE | LapPE | LapPE |
| batch size | 500 | 500 | 500 | 500 |
| #epochs | 150 | 150 | 150 | 150 |
| norm | - | - | - | LayerNorm |
| MPNN | - | - | - | GatedGCN |
| #Param. | 456k | 466k | 477k | 478k |

Table 7: Performance of our models on the Long-Range Graph Benchmark.

| METHOD | PASCALVOC-SP TEST F1 ↑ | COCO-SP TEST F1 ↑ | PEPTIDES-FUNC TEST AP ↑ | PEPTIDES-STRUCT TEST MAE ↓ | PCQM-CONTACT TEST MRR ↑ | | |
|---|---|---|---|---|---|---|---|
| | | | | | RAW | FILTER | EXT. FILTER |
| GCN | 0.2078 ± 0.0031 | 0.1338 ± 0.0007 | 0.6860 ± 0.0050 | 0.2460 ± 0.0007 | 0.3424 ± 0.0007 | 0.3631 ± 0.0006 | 0.4526 ± 0.0006 |
| GINE | 0.2718 ± 0.0054 | 0.2125 ± 0.0009 | 0.6621 ± 0.0067 | 0.2473 ± 0.0017 | 0.3509 ± 0.0006 | 0.3725 ± 0.0006 | 0.4617 ± 0.0005 |
| GATEDGCN | 0.3880 ± 0.0040 | 0.2922 ± 0.0018 | 0.6765 ± 0.0047 | 0.2477 ± 0.0009 | 0.3495 ± 0.0010 | 0.3714 ± 0.0010 | 0.4670 ± 0.0004 |
| GPS | 0.4440 ± 0.0065 | 0.3884 ± 0.0055 | 0.6534 ± 0.0091 | 0.2509 ± 0.0014 | 0.3498 ± 0.0005 | 0.3722 ± 0.0005 | 0.4703 ± 0.0014 |

## B    Additional Experiments

In Table 8 we provide extended results for the ablation study from Figure 1b. More specifically, we evaluate both GPS and GCN models on Peptides-Struct with predictive heads of various depths. As in Figure 1b, we study the isolated effect of this hyperparameter on the performance and leave all other hyperparameters identical to those used by Dwivedi et al. (2022).

The main observation is the large performance jump of GCN when configured with a head of depth two or three instead of the linear head that was originally chosen. The results of GPS do not suffer from a linear prediction head, but also do not improve further for deeper configurations. This is probably explained by the fact that each GPS layer can already process global graph information with a deep feed-forward network. A deeper prediction head may therefore be redundant in GPS models.

Table 8: Detailed performance on Peptides-Func for GPS and GCN with the depth of the prediction head varying from 1 to 3 layers.

| depth | GCN (MAE) | GPS (MAE) |
|---|---|---|
| 1 | $0.3496 \pm 0.0013$ | $0.2500 \pm 0.0005$ |
| 2 | $0.2547 \pm 0.0019$ | $0.2516 \pm 0.0012$ |
| 3 | $0.2534 \pm 0.0013$ | $0.2546 \pm 0.0020$ |

