# OpenReview forum: "Where Did the Gap Go? Reassessing the Long-Range Graph Benchmark"
_TMLR — Accepted by TMLR_

### Review · Reviewer_oXh1 · 2024-02-28

**Summary Of Contributions:**

This paper investigates the performance gaps in Graph transformer architectures compared to MPNNs on Long-Range Graph Benchmarks (LRGBs).
The authors observed that the performance gap is mostly due to suboptimal hyperparameter choices. In addition, the paper discusses the impact of lacking feature normalization on LRGB’s vision datasets and highlights the drawbacks of current LRGB’s link prediction metric.

**Audience:**

Yes

**Broader Impact Concerns:**

No ethical concerns have been found.

**Claims And Evidence:**

Yes

**Requested Changes:**

I kindly request that the authors address my concerns raised in the "Weaknesses" section. Once the authors respond, I will re-evaluate my assessment accordingly.

**Strengths And Weaknesses:**

Pros
1. The topic raised by the authors is very important for the graph learning community
2. The experiments and comparisons provided by the authors clearly show the importance of the author's study.

Cons.
1. The related work should be rewritten since it is not informative and does not answer how the covered methods tackle the problem of propagating the information from distant nodes for working on LRGBs.
2. Since authors argue that the reported performance gap is overestimated due to suboptimal hyperparameter choices in LRGB datasets, it is also essential to show other graph benchmarks, inductive/transductive and heterophilic, showing how the phenomena observed by the LRGBs generalize there.

---

> ### Comment · Reviewer_oXh1 · 2024-04-02
> **Request for authors feedback**
>
> Dear Authors,
>
> I would appreciate your responses to my comments. If you believe that the study you conducted is limited to LRGB datasets, I would like to hear your thoughts on whether the observed phenomena would generalize across other datasets.
>
> Regards,
>
> Reviewer oXh1

---

> > ### Author Response · Authors · 2024-04-02
> > **Author Response**
> >
> > Dear Reviewer,
> >
> > let us comment on the generalization of our findings to other datasets.
> >
> > The main improvements explored in the paper are fair hyperparameter tuning, feature normalization, and implementing evaluation metrics exactly as specified.
> > All of these practices naturally generalize to other datasets in the sense that they will improve experimental comparisons that did not already adhere to them.
> > However, since these are standard practices, most reference results on other datasets already incorporate them.
> > For example, the popular superpixel datasets MNIST and CIFAR are similar to PascalVOC-SP and COCO-SP but do have suitably normalized features.
> > Our specific experiment on the impact of feature normalization is therefore not applicable in this case.
> > The same argument applies to reported results with correctly implemented metrics and equal tuning.
> >
> > Regarding heterophilic datasets, a critical re-evaluation of heterophilic baseline results with fair hyperparameter tuning and dataset analysis was conducted by [1].
> > The main result is indeed similar to ours, as standard GNNs were observed to be on par with heterophily-specific models.
> > They do not test Graph Transformers with global attention in this study and training such models on their improved datasets with >10K nodes per graph would be very expensive and out of scope for the work presented here.
> >
> > Regarding other commonly used benchmark datasets, such as ZINC or OGB-MOLPCBA, Graph Transformers are not reported to outperform non-transformer baselines by margins as significant as those originally reported for LRGB.
> > For example, on the ZINC dataset, GPS [2] reports the exact same MAE of 0.07 for models with and without global attention.
> > The main result of an originally large performance gap not being observable after changing the setup towards more standard practices is therefore not directly applicable to these.
> >
> > A notable exception is the PCQM4M dataset [3], where Graph Transformers are reported to yield consistently better results.
> > Here, the baseline results can also be expected to be well-tuned, as they were obtained in a large public competition.
> > The success of transformers on this dataset may be due to its large size, which would align with observations in other fields, namely computer vision, where transformers require large-scale data to outperform alternative architectures [4].
> > However, evaluating this hypothesis for Graph Transformers is beyond the scope of this paper and other work is currently underway to test the scaling behavior of message passing GNNs and Graph Transformers to very large datasets [5].
> >
> > [1] Platonov, Oleg, et al. "A critical look at the evaluation of GNNs under heterophily: Are we really making progress?." arXiv preprint arXiv:2302.11640           (2023).
> >
> > [2] Rampášek, Ladislav, et al. "Recipe for a general, powerful, scalable graph transformer." Advances in Neural Information Processing Systems 35 (2022): 14501-14515.
> >
> > [3] Hu, Weihua, et al. "Ogb-lsc: A large-scale challenge for machine learning on graphs." arXiv preprint arXiv:2103.09430         (2021).
> >
> > [4] Dosovitskiy, Alexey, et al. "An image is worth 16x16 words: Transformers for image recognition at scale." arXiv preprint arXiv:2010.11929         (2020).
> >
> > [5] Beaini, Dominique, et al. "Towards foundational models for molecular learning on large-scale multi-task datasets." arXiv preprint arXiv:2310.04292       (2023).

---

### Review · Reviewer_9SYh · 2024-03-19

**Summary Of Contributions:**

This paper studies the empirical performance of message passing GNNs in comparison to Graph Transformers on the Long Range Graph benchmark. The principal findings demonstrate that after appropriate tuning, judicious choice of normalization, and filtering the gap between Graph Transformers and MPGNNs diminishes. The findings raise questions regarding the empirical rigor of past graph transformer  architectures.

**Audience:**

Yes

**Claims And Evidence:**

Yes

**Requested Changes:**

- Please give a bit more information about the important hyperparameters tuned to obtain the improved results for MPGNNs in the main text.

- Please also consider including a GAT and GIN architecture as baselines.

**Strengths And Weaknesses:**

This paper is a clear accept and it is important for broader community to immediately adopt its findings.

**Strengths**

This paper enjoys being simple and to the point. The message is quite effective and there is no extra fluff in the presentation which helps deliver the main message: carefully tuning and understanding your network through robust visualization is the key enabler of progress in deep learning. To this effect, this paper identifies key areas where the supposed supremacy of Graph Transformer architecture falls short. In particular, the issue with feature normalization in the vision dataset is an important finding. Furthermore, the finding about the MRR metric and the erroneous code piece that is probably now standard in many repositories is something very important to highlight to the broader community. I believe it is extremely important that evaluation metrics are free of these clear issues and as a result, this paper is a great contribution to this effort.

**Weaknesses**
There are no gross weaknesses in this work. The presentation can be slightly improved though. In particular, the hyperparameter subsection in section 2 felt like it could benefit from a deeper discussion.

---

> ### Author Response · Authors · 2024-03-22
> **Author Response**
>
> We thank the reviewer for the very positive review. We agree that the description of the hyperparameters was a little too concise and will add an additional discussion there (mentioning all hyperparameters that we tuned and that the depth of the head was surprisingly important).
>
> With respect to the baselines that we considered, we only use exactly the same MPGNNs that were also given in the original dataset paper.
> This choice of models allowed us to directly present the improvement, highlighting the importance of hyperparameter tuning.
> Note that the choice of MPGNNs already covers a representative range of architectures: simple convolutional GNNs (GCN), maximally expressive GNNs (GINE, which is GIN with edge features), and adaptive edge weights (GatedGCN).
> We would therefore stick to the concise comparison between these three MPGNNs and Graph Transformers.

---

### Review · Reviewer_Whiu · 2024-03-21

**Summary Of Contributions:**

This paper re-evaluates the performance of Message Passing GNNs and Graph Transformers on the Long-Range Graph Benchmark (LRGB). Through a rigorous empirical analysis, the authors find that the reported superiority of Graph Transformers is overestimated due to suboptimal hyperparameter choices. In fact, after basic hyperparameter optimization, the performance gap between the two model classes completely vanishes across multiple LRGB datasets. The main contribution of the paper is to highlight the importance of thorough empirical evaluation and establishing higher standards of rigor in graph machine learning research.

**Audience:**

Yes

**Claims And Evidence:**

Yes

**Requested Changes:**

Please refer to the weaknesses section.

**Strengths And Weaknesses:**

Strengths:

1. The paper addresses an important question regarding the performance of different models on graph learning tasks that heavily rely on long-range interactions.
2. The writing style of the paper is well-executed, ensuring clarity and coherence.

Weaknesses:

1. While I found the paper enjoyable to read, there are areas that could benefit from further improvement before publication. One such area is the lack of sufficient background introduction, particularly for readers who are not familiar with the LRGB benchmark. Providing more context would enhance the readability for a broader audience.
2. It would be valuable to include additional experiments to support the findings. While it is expected that tuning hyperparameters can improve GNN performance, it would be interesting to explore if Graph Transformers exhibit more robust performance across different hyper-parameters.

---

> ### Author Response · Authors · 2024-03-22
> **Author Response**
>
> We thank the reviewer for the positive review.
>
> We agree that the introduction of LRGB could be more detailed, especially for readers who are unfamiliar with the dataset. This would indeed help the presentation. We will expand this and add the main reasons why the LRGB datasets are conjectured to be more long-range than previous datasets (i.e. bigger graphs, dependence on whole-scene statistics, and manually enforcing a minimum distance in PCQM-Contact).
>
> In our experiments, GPS was not more stable during hyperparamater tuning compared to GCN and the other two MPGNNs. In particular, it was of utmost importance when tuning GPS to choose the right embedding (i.e. RWSE or LapPE) and normalization which both were dataset dependent. We will discuss this in more detail and add a plot to the appendix that shows the spread of the GPS and GCN results across tuning runs.

---

### Review · Reviewer_vdfV · 2024-03-23

**Summary Of Contributions:**

This paper target the phenomenon 'graph transformers significantly outperform mpgnns' claimed in the LRGB work, and revaluate different MPGNNs and graph transformers. The first conclusion is that the performance gap in LRGB is overestimated because of the suboptimal hyperparameter choice. Second, feature normalization is founda to be beneficial. Finally, different filtering strategies have a majot impact on the results.

**Audience:**

Yes

**Broader Impact Concerns:**

No ethical concern

**Claims And Evidence:**

No

**Requested Changes:**

Question:

Although reevaluating the experiments by other researchers is beneficial, how to justify that the experiments in this paper is perfectly rigorous? The search space of the hyperparameters is infinite, and the authors cannot prove that the hyperparameters found in this paper is a global optimum. Therefore, it is still possible that other researchers also reevaluate the experiments in this work, and find hyperparameters leading to completely contradictory conclusions.

**Strengths And Weaknesses:**

Strengths:

The rigor of experiments is very important for the machine learning community, and the paper makes contribution in this perspective.

Weakness:

This paper is mainly adjusting the hyperparameters, the normalization, and metric of the models, but does not provide technical or theoretical contribution to machine learning or representation learning. Therefore, I'm not sure whether TMLR is a suitable venue for it.

---

> ### Comment · Reviewer_9SYh · 2024-03-23
> **Ridiculous review**
>
> Hi,
>
> As a fellow reviewer I am deeply concerned by the narrative posed in this review. The claim that the search space of hyperparameters is infinite and that one cannot prove that there is a global optimum is a ludicrous claim. I think it's absolutely fine that the authors pinpoint the exact failure modes of past evaluations. To dismiss the contribution that the MRR evaluation has been broken and the authors actually fix this as a hyperparameter search is disingenuous.
>
> The fact that findings poke a hole in the entire subfield of Graph Transformers is extremely worthy of TMLR. We do not need novelty here to advance the field. I encourage the reviewer to re-read the paper and provide more constructive feedback to the authors if needed.

---

> > ### Comment · Reviewer_vdfV · 2024-03-25
> > **Clarification**
> >
> > Thanks for raising the concern, and I believe it is necessary to clarify my comments, which may be helpful to the auhors.
> >
> > **1.** As stated in Summary of Contributions, this paper made contribution in (i) filling the performance gap between MPGNN and graph transformer by hyperparameter-tuning, (ii) Strengthening the importance of feature normalization (on two CV datasets), and (iii)
> > pointing out the importance of the filtering strategies used in MRR evaluation metric.
> >
> > As acknowledged in Strengths, this contribution towards the rigor of experiments is 'very important'.
> >
> > Therefore, I hold a positive attitude to the main content and contribution of the paper.
> >
> > **2.** Hyperparameter search space. Regarding this, the authors did not conduct dense grid search, and each hyperparameter got at most 3 candidate values. This is fine, since it is intractable to conduct dense hyperparameter search due to the large amount of possible combinations.
> >
> > But a natural question rises here: In this paper, the tested hyperparameters can improve MPGNN performance, but does not improve the GPS performance at all. Accordingly, the conclusion is that graph transformers do not possess the superiority over MPGNNs claimed in previous research. Then, is it possible that some other hyperparameter combinations can further improve the performance of graph transformers (e.g. GPS) and reach the oppsite conclusion?
> >
> > As specially noted in my comments, this is a 'question' instead of a 'weakness'. I would like to discuss it with the authors because it is not only revelant to this work but also relevant to all machine learning research areas. Given two machine learning models with different structures, how should we judge which is better to a target task? Besides the potential unreached hyperparameter space, there are many other complicated factors. For example, this paper and the original LRGB paper adopt the setting of fixing the parameter amount to ensure the fairness.
> > But the optimal parameter amount may not be same for different models, and how to ensure this pre-defined parameter amount is proper? E.g. the original GCN model without residual connection experiences over-smoothing when going deeper than 3 or 4 layers, while other models may require more parameters to reach the peak performance.
> >
> > Above all, I think the question on comparing different model structures is not that simple. And that is why I did not regard this as a 'weakness' of the paper, but only seek to discuss it.
> >
> > Since 'establishing a higher standard of empirical rigor' is the principle aim of this paper, I think this question is highly relevant to this paper.
> >
> > **3.** Whether the paper fits TMLR. As stated in my comments, I think this is questionable, but I'm not sure about it. Re-evaluation paper is not common, so I think it is necessary to note it down in review to ensure the editor is aware of this issue.

---

> ### Author Response · Authors · 2024-03-25
> **Author Response**
>
> We thank the reviewer for the feedback and especially the clarifications in the comments.
>
> We would like to comment especially on the question on hyperparameters.
> The hyperparameter ranges we use are in line with common practice and identical for all methods.
> For all compared methods the validation performance also saturated towards the end of each tuning sweep, showing that the hyperparameter ranges are sufficiently large to converge to a representative, although not globally optimal, configuration.
> Additional tuning may increase performance further, but with diminishing returns - both for MPGNNs and GPS.
> The main result then is that the original large reported advantage of graph transformers is not observable after performing a standard hyperparameter sweep for all methods.
> We would also like to point out that our changes did improve GPS on 3 out of 5 datasets, although in the CV datasets feature normalization was a key factor too.
>
> We also note that follow-up graph transformers tuned by their respective authors are not massively better than GPS or our MPGNN results on the peptides datasets.
> This indicates, that there is indeed no big conceptual gap between GTs and MPGNNs on these datasets.
> However, the optimality of these newer results may also be questioned since few papers report details on how they tune configurations.
>
> Of course, the exact tuning methodology is a non-trivial design choice with no one-fits-all solution.
> Since any tractable experimental setup has to make some compromise regarding which configurations to consider it is crucial to publicly report the exact tuning methodology.
> Unfortunately, many studies neglect this aspect and do not report these details, which ultimately yields unreliable comparisons in follow-up studies.
> A key contribution of this work is to provide updated baseline results for LRGB while also specifying exactly how the model configurations were tuned.
> Authors who use these numbers as baselines in future research can therefore directly check whether their hyperparameter budgets are similar to ours and if the comparison is fair.
>
>
> The fixed number of parameters is indeed a fairness constraint that not all researchers may agree to.
> It is very common in graph learning and other fields (especially NLP) to prioritize comparisons between models of equal size.
> This constraint is not perfect, as for example a GCN and a GPS model with an identical number of parameters have quite different runtimes with the GCN being a lot faster.
> In order to compensate for that, one could instead constrain the wall-clock runtime on identical hardware.
> Since the creators of the LRGB dataset suggested a comparison with a parameter budget, we stuck to this constraint for simplicity.
> Overall, we believe that the question of how to compare architectures in a fair manner is indeed relevant to the paper and we will add a short discussion of this topic.
>
>
> We argue that our results are a significant technical contribution to the field of graph learning where the design of new methods, such as graph transformers, is heavily guided by empirical experimentation.
> To aid further research, we provide improved baseline results alongside hyperparameter budgets and outline pitfalls (such as the importance of feature normalization or the right choice of link prediction metric) when working on some of the most popular benchmark datasets that are currently used.

---

> > ### Comment · Reviewer_vdfV · 2024-03-30
> > **Thanks for the detailed response**
> >
> > Thanks for the detailed responses from the authors. The promised additional discussion on how to fairly compare different architecture will further improve the value of the paper by reminding more researchers of this critical problem.
> >
> > Most parts of the response are clear, but I'm a little bit confused by one thing. Could the authors please explain more on 'the validation performance saturated towards the end of each tuning sweep, showing that the hyperparameter ranges are sufficiently large to converge to a representative configuration'? Does 'each tuning sweep' refers to choosing different values for each hyperparameter? By 'towards the end of each tuning sweep', does it mean to tune the hyparameter towards a certain direction, e.g. increase from 0.1 to 0.5?

---

> > > ### Author Response · Authors · 2024-04-02
> > > **Author Response**
> > >
> > > We apologize for the vague phrasing.
> > > With this, we simply mean to say that expanding the hyperparameter budgets further would only yield marginal gains.
> > > With 'each tuning sweep' we refer to the whole sweep across all tested hyperparameter combinations, which is not done in any particular order.
> > >
> > > Let us know if any further clarifications are needed.

---

### Decision · Action_Editor_q2Y8 · 2024-05-01

**Recommendation:** Accept with minor revision

**Comment:**

The authors investigate the empirical performance of message passing GNNs in comparison to Graph Transformers on the Long Range Graph benchmark. The principal findings demonstrate that after appropriate tuning, judicious choice of normalization, and filtering the gap between Graph Transformers and MPGNNs diminishes. The findings raise questions regarding the empirical rigor of past graph transformer architectures.

The topic raised by the authors is very important for the graph learning community. Specifically, the authors aim to address an important question regarding the performance of different models on graph learning tasks that heavily rely on long-range interactions. The experiments and comparisons provided by the authors clearly show the importance of the author's study. The writing style of the paper is well-executed, ensuring clarity and coherence. However, this paper is mainly adjusting the hyperparameters, the normalization, and metric of the models, but does not provide technical or theoretical contribution to machine learning or representation learning. Moreover, this paper lacks the sufficient background introduction, particularly for readers who are not familiar with the LRGB benchmark. It would be valuable to include additional experiments to support the findings. While it is expected that tuning hyperparameters can improve GNN performance, it would be interesting to explore if Graph Transformers exhibit more robust performance across different hyper-parameters. Therefore, based on four qualified reviews, this paper can be accepted with minor revision and the authors are encouraged to merge the comments into their update versions.

**Audience:**

Yes

**Claims And Evidence:**

Yes